# Targeting Myeloid Differentiation Primary Response Protein 88 (MyD88) and Galectin-3 to Develop Broad-Spectrum Host-Mediated Therapeutics against SARS-CoV-2

**DOI:** 10.3390/ijms25158421

**Published:** 2024-08-01

**Authors:** Kamal U. Saikh, Khairul Anam, Halima Sultana, Rakin Ahmed, Simran Kumar, Sanjay Srinivasan, Hafiz Ahmed

**Affiliations:** GlycoMantra Inc., bwtech South of the University of Maryland Baltimore County, 1450 South Rolling Road, Baltimore, MD 21227, USA; khairulanamindia@gmail.com (K.A.); hsultana@glycomantra.com (H.S.); rakinahmed95@gmail.com (R.A.); simrankr0103@gmail.com (S.K.); sanjay@glycomantra.com (S.S.)

**Keywords:** MyD88, galectin-3, COVID-19, SARS-CoV-2, IFNs, cytokine, TLRs, NLRP3, PRR

## Abstract

Nearly six million people worldwide have died from the coronavirus disease (COVID-19) outbreak caused by severe acute respiratory syndrome coronavirus 2 (SARS-CoV-2) infection. Although COVID-19 vaccines are largely successful in reducing the severity of the disease and deaths, the decline in vaccine-induced immunity over time and the continuing emergence of new viral variants or mutations underscore the need for an alternative strategy for developing broad-spectrum host-mediated therapeutics against SARS-CoV-2. A key feature of severe COVID-19 is dysregulated innate immune signaling, culminating in a high expression of numerous pro-inflammatory cytokines and chemokines and a lack of antiviral interferons (IFNs), particularly type I (alpha and beta) and type III (lambda). As a natural host defense, the myeloid differentiation primary response protein, MyD88, plays pivotal roles in innate and acquired immune responses via the signal transduction pathways of Toll-like receptors (TLRs), a type of pathogen recognition receptors (PRRs). However, recent studies have highlighted that infection with viruses upregulates MyD88 expression and impairs the host antiviral response by negatively regulating type I IFN. Galectin-3 (Gal3), another key player in viral infections, has been shown to modulate the host immune response by regulating viral entry and activating TLRs, the NLRP3 inflammasome, and NF-κB, resulting in the release of pro-inflammatory cytokines and contributing to the overall inflammatory response, the so-called “cytokine storm”. These studies suggest that the specific inhibition of MyD88 and Gal3 could be a promising therapy for COVID-19. This review presents future directions for MyD88- and Gal3-targeted antiviral drug discovery, highlighting the potential to restore host immunity in SARS-CoV-2 infections.

## 1. Introduction

COVID-19 is a consequence of SARS-CoV-2 infection, which was first reported in China in late 2019, resulting in nearly six million deaths worldwide [1,2]. Although COVID-19 vaccines are largely successful in reducing the severity of the disease and deaths, vaccine-induced immunity declines over time. Moreover, new virus variants or mutations are continually emerging, posing a threat to global public health, which underscores the need for a new strategy to develop broad-spectrum host-mediated therapeutics against viral infections, including SARS-CoV-2. A key feature of severe COVID-19 is dysregulated innate immune signaling, culminating in a high level of expression of numerous pro-inflammatory cytokines and chemokines, such as IL-6, TNF-α, and IL-1β—the so-called “cytokine storm”—as well as a lack of response from antiviral type I (α and β) and type III (λ) interferons (IFNs) (known to be the most potent natural antiviral mediators) [3,4]. 

However, the lack of adequate type I and III interferon responses and their potential roles in controlling SARS-CoV-2 viral replication have not been fully addressed, although some studies have suggested that a dysregulated and delayed host interferon response to SARS-CoV-2 virus infection contributes to a persistent viral presence and disease progression [5,6]. Given what is known about the dysregulated host innate response to COVID-19, in a future scenario of a COVID-19-like viral pandemic, a rational approach to a drug discovery platform would be to consider targeting a combination of host factor(s) at the cell surface level, preventing viral entry and shielding the intracellular host factor(s) critically involved in regulating the immune signaling pathways that are crucial in balancing host-mediated immunity.

Host exposure to microbial and viral pathogens or pathogen-associated molecular patterns (PAMPs) is generally recognized by a set of pathogen recognition receptors (PRRs), including Toll-like receptors (TLRs) [7,8,9]. The binding of PAMPs to TLR(s) initiates the activation of innate immune signaling cascades, leading to the induction of inflammatory responses, which later progress to antigen-driven precise adaptive immunity. Besides TLR3, most TLR-initiated inflammatory responses are primarily mediated by the involvement of a signaling adaptor protein, the myeloid differentiation primary response protein 88 (MyD88) [10,11]. MyD88 is a cytosolic anchor adaptor protein that plays an essential role in inducing the innate and acquired immune responses that are driven by TLRs and receptors of the interleukins IL-1 and IL-18 (IL-1R and IL-18R) [12,13].

In addition, a β-galactoside-binding lectin, galectin-3 (Gal3), has emerged as a pivotal player in host–pathogen interactions and viral infections [14]. Gal3 is believed to function as both a PRR and a danger-associated molecular pattern (DAMP) [15,16,17]. In the context of SARS-CoV-2 infection, Gal3 has been shown to modulate the host immune response through the regulation of viral entry and the activation of TLRs, the NLRP3 inflammasome, and NF-κB, resulting in the release of pro-inflammatory cytokines and contributing to the overall inflammatory response, the so-called “cytokine storm” [14]. This review presents future directions for MyD88- and Gal3-targeted antiviral drug discovery, highlighting the role of SARS-CoV-2 or any future COVID-19-like emergent viral diseases in balancing host immunity.

## 2. MyD88

The human MyD88 gene contains five exons and four introns, spanning 4.54 Kb. They have an open reading frame of ~2.6 kb, which translates into a protein of approximately 33 kDa (296 amino acids) [18,19] (Figure 1). The MyD88 protein consists of three main domains: an N-terminal death domain (DD); a Toll-interleukin-1 receptor (TIR) domain at the C-terminus; and an intermediary domain (ID), which separates both IRAK and TIR [11,20] (Figure 1). 

The C-terminal TIR domain of MyD88 binds to the TIR domain of the receptor [22], and the N-terminal DD is involved in binding to IL-1R-associated kinase (IRAK) 4 for further signaling. The INT domain is also essential in signaling [23]. The structure of MyD88 is unique, as its TIR domain is involved in both heterodimerization with the TIR domain of other receptors such as TLR-IL-R1 and homodimerization with another MyD88 molecule for the recruitment of downstream signaling molecules (Figure 2). The exposed BB loop (aa196-202) in the TIR domain is critical for TIR–TIR interactions and the MyD88-mediated inflammatory response [24].

In general, viral infection or viral products, such as double-stranded RNA (dsRNA) produced during viral replication, are recognized by TLRs and RIG-I/MDA 5, which trigger two key immune signaling pathways in which interferon regulatory factors (IRFs) and NF-κB are critical elements. IRFs are crucial transcription factors for the expression of IFN, leading to the first wave of the host cell response, and NF-κB is known to regulate pro-inflammatory cytokines and type I IFNs during infection. The left box presents a schematic representation, showing MyD88 heterodimerization with the TIR domain of the receptor and homodimerization with another molecule of MyD88. MyD88 dimer formation and its recruitment to the signaling cascade lead to the upregulation of NF-κB-mediated pro-inflammatory cytokines during viral infection. A synthetic compound named compound 4210, designed based on the model of the tripeptide sequence of the BB loop in the TIR sequence, can interfere with the dimerization of MyD88 inhibitors and downregulate pro-inflammatory cytokines (for details, see Ref. [11]). The right box shows that, upon sensing pathogens or PAMPs, host innate immune receptors, PRRs such as TLRs, and retinoic acid-inducible gene-I (RIG-I)-like receptors (RLRs)/melanoma differentiation-associated gene-5 (MDA-5) activate innate immune signaling cascades through the recruitment of specific adaptor proteins, such as mitochondrial antiviral signaling protein (MAVS), TANK-binding kinase (TBK1), and IkB kinase (IKK). During viral infection, MyD88 is upregulated, and the interaction of MyD88 through its TIR domain with IRF-3/IRF-7 sequesters IRF-3/IRF-7. As shown in the schematic representation, a synthetic dimeric MyD88 inhibitor, compound 4210, binds to the exposed BB loop of MyD88 [11,25], thereby preventing the interaction of MyD88 with IRFs, and it interferes with the sequestering ability of MyD88 and allows for the full activation of IRF-3 and IRF-7, including phosphorylation, dimerization, and translocation to the nucleus for a type I IFN response (adapted from Ref. [11]; TRIF, TIR-domain-containing adaptor-inducing IFN-β). The SARS-CoV-2 image was taken from the Centers for Disease Control and Prevention (CDC) (https://stacks.cdc.gov/view/cdc/86942, accessed on 21 May 2024).

The host induction of the innate immune response at the onset of infection is mediated and controlled by intracellular signaling cascades via the recruitment of MyD88 and its association with the MyD88 adaptor-like (Mal, also known as TIR-containing adaptor protein or TIRAP). This association leads to the activation of downstream pro-inflammatory signaling cascades [26]. The activation of host innate immunity is strongly dependent on the tightly regulated function of MyD88. The dysregulation or uncontrolled activation of MyD88 may cause an imbalance in the host immune response, leading to a wide range of inflammation-associated syndromes and pathogeneses. In many viral infections, the upregulation of MyD88 is associated with a decreased antiviral type I IFN response; however, MyD88-deficient mice have shown an increased type I IFN response and survivability. For example, infections with Coxsackie virus B3, Venezuelan equine encephalitis virus (VEEV), or Marburg virus significantly increased MyD88 [27,28,29]. An increase in MyD88 in the cytosol has been proposed to exert an inhibitory effect through MyD88 and IRF3/IRF7 interactions and limit IRF availability, thereby curtailing the type l IFN response. Consistent with these reports, it has been demonstrated that, following poly I:C (dsRNA) stimulation, IFN-beta gene induction significantly increases MyD88 in Mal/TIRAP-deficient cells and in wild-type cells treated with a Mal inhibitory peptide [30,31]. Reports suggest that, while MyD88 upregulation is required for the effective activation of the host innate immune response, MyD88 upregulation through the interaction of multiple interferon regulatory factors such as IRF3 and IRF7 contributes negatively, with this being initiated via other pathways of innate immune signaling such as TLR3 and RIG-I/MDA, resulting in a decreased type I IFN response and allowing for virus spread and disease progression [11]. Because of the expanding role of anchor adaptor proteins in controlling the host immune response, MyD88 has emerged as an attractive drug target in restoring the host-mediated immune response. Based on the BB loop’s conserved amino acid sequence in the TIR domain of MyD88, a structure-based approach has been utilized to design small-molecule inhibitors of MyD88. For example, we developed a synthetic small-molecule MyD88 inhibitor (compound 4210), which blocked TIR–TIR domain homodimerization and showed therapeutic efficacy in attenuating the MyD88-mediated inflammatory impact and increasing the antiviral type I IFN response in an experimental mouse model of diseases [11,21,25].

### 2.1. Role of MyD88 in the Regulation of Host Immunity in Viral Infections—“Host Friendly” or “Hostile”?

The innate and subsequent adaptive immune responses to viral infections, including SARS-CoV-2, are generally initiated at the cellular level after viral evasion [32]. After viral entry, the infected cell recognizes the presence of viral replication using any PRR. The physical engagement of these distinct structures (PRRs), such as TLRs/IL-R via MyD88-TRIF, the RIG-I/MDA-5–MAVS axis, double-stranded RNA-dependent protein kinase (PKR), the DNA receptor, DAI, and the cyclic GMP-AMP synthase (cGAS) stimulator of interferon genes (STING) axis for cytosolic RNA and DNA, respectively, facilitates the recognition of different pathogens; thus, they serve as sentinels for various microbes inside and outside of the cell [33,34]. Double-stranded RNA, an intermediate byproduct of viral replication, is also detected by intracellular PRRs [33], and the interaction of many virus-specific RNAs culminates in receptor oligomerization and begins a concerted activation of networks of innate immune signaling pathways and cellular immune responses against the invading pathogens.

Briefly, in infected cells, a signaling chain is activated upon the detection of viral double-stranded RNA (dsRNA) binding to RIG-I or MDA5, which are ubiquitously expressed in most tissues and appear to function in parallel with varying degrees of virus specificity. RIG-I- or MDA5-mediated signaling for the interferon response occurs after binding to downstream factors, called interferon beta promoter stimulator-1 (IPS-1) or MAVS; this leads to the activation of IRF3 kinases, such as TBK-1 or IKKε, which are known to phosphorylate IRF-3 or IRF-7. Phosphorylated IRF homodimerizes and translocates into the nucleus, where it recruits the transcriptional co-activators p300 and CREB-binding protein (CBP) to initiate IFN-β mRNA expression. NF-κB and AP-1 are also recruited in a dsRNA-dependent way. As described earlier, the engagement of TLR3 using dsRNA triggers downstream signaling via the adaptor molecule TRIF, which bypasses IPS-1/MAVS and directly activates kinase TBK-1 and, subsequently, the IFN response, as described earlier. Hence, these transcription factors strongly trigger the “first wave” of the IFN response.

In general, the initial activation of the antiviral innate immune response to viral infections primarily depends on the expression of antiviral cytokines, such as type I and III IFNs, which are produced by various cells [35,36,37] through the activation of downstream transcription factors, such as IRFs and NF-kB, via MyD88-dependent and -independent immune signaling pathways [38]. The transcriptional activation of IRFs and NF-kB upregulates IFN-stimulated genes (ISGs) and antiviral proteins, such as dsRNA-activated protein kinase R, 2-5 oligo-adenylate-synthase, and M × 1, ultimately mediating the antiviral actions of IFN. Because of these strong host-directed innate immune responses, many viral infections are generally resolved without further complications. To constitute a prolific infection, viruses must escape and overcome these initial antiviral type I (IFN α/β) and type III IFN (IFN-λ, 1-IV) responses. A lack of or the delayed host cell synthesis of IFNs in sufficient quantities is a cause of failure in mounting a robust antiviral response [39]. Type I IFNs also enhance the adaptive immune response through the activation of other cells of the immune system, such as T cells, B cells, dendritic cells, and NK cells, and they thwart pathogen dissemination and prevent disease progression [40]. Thus, as a first line of defense against viral infections, the general perspective of immunity is that the interferon-mediated antiviral immune response, which precedes pro-inflammatory responses, optimizes host protection in balancing immunity and maintains collateral damage.

Many reports suggest that, for COVID-19 disease, the above paradigm does not apply [41]. In particular, the untuned antiviral response observed in COVID-19 revealed that both type I and type III IFNs were reduced or delayed, being induced only in a few patients. In contrast, exacerbated pro-inflammatory cytokines, such as TNF-α, IL-6, and IL-8, were produced for a long time prior to IFN synthesis in all patients, contributing to persistent viral presence, hyperinflammation, and respiratory failure. Hatton et al. [5] reported that an untuned or delayed induction of antiviral type I and III IFN responses in COVID-19 mediates the permissiveness of nasal epithelial cells to SARS-CoV-2. However, when provided prior to infection, recombinant IFN β or IFN λ 1 can efficiently induce an antiviral condition that preserves epithelial barrier integrity and potentially restricts SARS-CoV-2 spread [5]. More recently, Viox et al. reported that a mutated form of IFNalpha 2, termed IFNmod, which can elicit weak IFN-I signaling, and the resultant type I IFN response potentially inhibit SARS-CoV-2 replication and inflammation, with only moderate disease being observed in experimental rhesus macaques [6]. Due to its significant role in antiviral defense, type I IFN has also been recommended for the recent COVID-19 pandemic [42]. 

Generally, in many viral infections, transcription factors, particularly IRFs, control the expression of IFN, and pro-inflammatory cytokines are regulated by the activation of NF-κB and type 1 IFNs. Other cellular factors, such as MAVS, TRIF, TRAF3/TRAF6, and MyD88, and downstream kinases, such as IKK-e/TBK and IKB (but upstream of IRFs and NF-κB), can facilitate the transcriptional activation of IRFs and NF-kB. It is also important to note that dsRNAs are recognized by more than one PRR, such as TLRs and RLRs/MDA-5, wherein the activation of IRF-3/IRF-7 is essential for regulating the induction of IFN-α/β [43,44,45,46]. At the onset of viral infection, the produced type I IFN interacts with the interferon receptor (IFNR). The binding of IFN α/β to the IFNR subsequently activates the JAK1-STAT pathway, leading to the assembly of IFN-stimulated gene factor 3 (ISGF3). The ISGF3 complex, which consists of STAT1-STAT2 dimers and IRF9, binds to IFN-stimulated response elements (ISREs) in the promoters of IFN-stimulated genes (ISGs) to provide a robust expression of the second wave of various ISG genes. This fully fledged activation of host innate immunity through autocrine and paracrine signaling in surrounding uninfected cells creates an antiviral state. This can be achieved without the constraints or limitations of IRF-3/IRF-7, which is required for IRF phosphorylation, a critical element in the first wave of strong IFN-I induction [30,31,43,47,48]. The intracellular sequestration of IRFs would limit the required IRF phosphorylation. It would result in weak IFN-I signaling and a deficiency in IFN-I output, which is needed for robust antiviral host immunity [30,31].

Thus, in the TLR immune signaling pathway that induces the type I IFN response, MyD88 is used by all TLRs, except for TLR3, and it is shared by IL-1, IL-18, and IL-33 receptors, while TLR3 and TLR4 exclusively engage TRIF. Further, while the recruitment and activation of MyD88 and TRIF are critical for a robust innate immune response, these two proteins have been shown to be activated after viral infection [49] and often exert an opposing effect on the expression of inflammatory genes [50]. It has been found that the effects of TLR adaptors, namely, MyD88 and MAL/TIRAP, are involved in the negative regulation of alternative TLRs [30,31,51,52]. While MyD88 is known to activate all TLRs, except for TLR3, it has also been found that MyD88 functions negatively in the TLR3-initiated immune signaling pathway. Besides TLR3, studies suggest that RIG-I and MDA-5 can detect RNA viruses or analogs (poly I: C), resulting in antiviral IFN-β induction via the activation of IRF-3 and IRF-7 [53]. Thus, TLRs and RLRs can work together or independently through the recruitment of MyD88 or TRIF in the perpetuation of downstream signaling networks to generate robust immune responses [53]. Yet, during many viral infections, such as Coxsackie virus B3, Venezuelan equine encephalitis virus (VEEV), or Marburg virus infection, the increase in MyD88 upregulation and MyD88-IRF3/IRF7 interaction leads to the sequestering of IRF3/IRF7 (limiting the availability of IRF3/IRF7), thus exerting an inhibitory effect on the TLR3- and TRIF-mediated downstream signaling of the type I IFN response. In this way, MyD88 interaction with IRF3/IRF7 in a MyD88-independent immune signaling pathway exerts weak immune signaling and curtails IFN-β induction, which is critical for clearing infection at an early stage [25,30,31]. As a single infectious agent may harbor multiple PRR agonists, which can trigger various PRRs, such as multiple TLRs, RIG-I and MDA 5 initiate innate immune signaling such as the TLR3-TRIF-IRF3/IRF7 or RIG-I/MDA5-MAVS axis (MyD88-independent signaling) for antiviral immune responses. The upregulation of MyD88 and interaction with IRF3 or IRF7 limit the availability of IRFs (sequester) and restrict fully fledged IRF phosphorylation, dimerization, and translocation to the nucleus for IFN synthesis.

These views are supported by results showing that IFN-β gene induction in MyD88- and Mal/TIRAP-deficient cells was significantly increased following poly I:C (dsRNA) stimulation or when wild-type cells were treated with a Mal/TIRAP-inhibitory peptide [30,31]. These results strongly support the view that the significant upregulation of MyD88 and Mal/TIRAP also negatively regulates IFN-β induction [30,31]. In vivo data evidently support this notion. MyD88^−/−^ mice were shown to have a significantly higher survival rate (86%) than MyD88^+/+^ mice (35%) after CVB3 or HSV-1 infection [27,47]. In addition, compared with MyD88^+/+^ DCs, a significant increase in IFN α/β, IRF1, IRF7, and ISGs was also observed in MyD88^−/−^ DCs following exposure to EBOV virus-like particles (eVLPs) [54]. Thus, MyD88 upregulation impaired the type I IFN response during many viral infections. In a mouse model of viral diseases, we showed that a small-molecule inhibitor of MyD88, compound 4210, limited the sequestration of IRF3/IRF7, resulting in increased IRF phosphorylation and type I IFN response; suppressed viral replication; and improved survival, weight change, and clinical disease scores [25] (see Figure 2). Human U87 cells, when stimulated with multiple TLR ligands in the presence of a MyD88 inhibitor (compound 4210), concurrently increased the phosphorylation of IRF-3, which was consistent with an increase in the type I IFN response. These results indicate an alternate TRIF-IRF3 axis-mediated IFN-β induction [55].

Apart from the acquired immune response, host cells initiate strong defenses through the innate immune response to viral infection. In many instances, viral infections occur without any significant consequences. Patients recover with virus elimination or incorporation into a latent or persistent form without subsequent problems. An increase in IFN is one of the most crucial events in this innate immune defense mechanism, and it acts as a primary switch for initiating antiviral host immunity. Thus, IFN plays a pivotal role in antiviral host defense against all types of viruses [11,25]. However, in this context, it is also important to note that, during viral infection, a combination of host factor(s), including MyD88, contributes to the innate immune signaling pathways, which can lead to beneficial (friendly) or detrimental (more like friendly fire) outcomes of the IFN response. In a normal immune activation process, type I IFNs (α and β) are known to have potent antiviral activities and broad functions and effects, both direct (i.e., mediating resistance to viral replication) and indirect (i.e., immune stimulation). Due to the significance of IFN in the host cell in restricting viral infections, particularly in the absence of an effective antiviral or vaccination strategy, IFN has been clinically approved (e.g., for the treatment of chronic hepatitis B or C).

### 2.2. MyD88-Targeted Therapeutic Approach to Tackle Exuberant Inflammation and to Augment Antiviral IFN Response to Viral Infections, Including SARS-CoV-2

The positive outcomes of early effective innate immune responses are largely dictated by host immunity and the activation of various immune cells. With the occurrence of a lack of or imbalance (dysregulation) in the immune control mechanism in host cells, viral immune evasion may lead to pathogenesis and disease progression. In many instances, a weak and delayed production of antiviral type I and III IFNs and an exuberant release of pro-inflammatory cytokines contribute to severe forms of viral disease pathology, including COVID-19 [56,57]. COVID-19 patients exhibit markedly elevated levels of pro-inflammatory cytokines (such as IL-6) and chemokines [58,59], which are known to be associated with severe pathology and impaired lung function. Although the dysregulation/overactivation of MyD88 is known to contribute to an exacerbated inflammatory response, the so-called “cytokine storm”, in COVID-19 patients, it is at least partly controlled by innate immune signaling. It is yet to be determined whether the upregulation of MyD88 is linked to the initial impairment of the type I IFN response in SARS-CoV-2 infection. It is known that the key components of innate immune signaling networks, including MyD88, contribute to an exacerbated inflammatory response [25]. In many viral infections, the upregulation of MyD88 impaired the type I IFN response, and the inhibition of MyD88 improved the type I IFN response, suppressed viral replication, and improved animal survival [25,27,60,61]. In general, following viral entry at the cell surface, and within the first few hours of viral replication, the induction and magnitude of the host innate immune responses, including type I and III IFN responses, determine the ability of the virus to spread and its subsequent effects. The untuned antiviral immunity associated with SARS-CoV-2 indicates that type I and III interferons are notably reduced and delayed in most COVID-19 patients [41]. In many cases, viral proteins are known to interfere with the induction of type I IFN and evade host antiviral innate immune effector mechanisms; it is yet to be determined whether IFN antagonists can be found in SARS-CoV-2 [56]. Studies have suggested that TLR adaptor signaling generates a balanced protective immune response to highly pathogenic SARS coronavirus infections, and TRIF-mediated immune signaling in particular has been shown to contribute to the protective innate immune response [57]. It is not yet known whether the upregulation of intracellular MyD88 is responsible for pro-inflammatory immune signaling, resulting in severe inflammation while reducing the antiviral type I IFN response through the sequestration of IRFs via TRIF pathways. In vertebrates, strong type I IFN induction at the onset of infection is a prerequisite to controlling viral infections through the modulation of the overall innate immune response, promoting antigen presentation in a balanced manner and enhancing natural killer cell functions [62]. In COVID-19 patients, an impaired T-cell response has been reported, leading to lymphopenia and the functional exhaustion of CD4^+^ and CD8^+^ T cells [63]. This may be a consequence of deficient IFN production. IFNs can also regulate the development of regulatory T cells (T_reg_) cells. In COVID-19 patients, T_reg_ cell counts are known to be inversely correlated with disease severity [58,64]. As the MyD88-mediated overactivation of pro-inflammatory signaling leads to a cytokine storm and as the production of antiviral type I IFNs is reportedly blunted, the therapeutic targeting of MyD88 may be highly advantageous in limiting severe inflammation and inducing vigorous TRIF-mediated immune signaling to augment the antiviral type I IFN response. Using high-throughput screening, small molecules with type 1 IFN-inducing properties have been identified; however, the molecular target and mechanism associated with IFN induction are not yet known [65]. We have reported a synthetic inhibitor of MyD88 (compound 4210) that functions as an immune-modulating molecule via the deactivation of MyD88, thereby promoting type I and III IFN signaling, which is central to IRF activation and a consequent strong host antiviral IFN response [19]. Similar to compound 4210, other small-molecule inhibitors of MyD88, such as T6167923 and S5 [21,66], have also been shown to induce type I IFNs (unpublished data). It is likely that an imbalanced and delayed type I IFN response in the host to SARS-CoV-2 drives the development of the severity of COVID-19 disease [59]. Therefore, it is anticipated that the pharmacologic inhibition of MyD88 may induce antiviral type I IFNs and provide a TIR blockade to limit pro-inflammatory cytokines. Compound 4210 has not yet been tested in a COVID-19 model; it is tempting to speculate that the combination of these two-pronged antiviral mechanisms is feasible for the potential therapy of COVID-19 or COVID-like diseases in balancing host immunity.

## 3. Galectin-3 

Galectin-3 (Gal3), a member of the β-galactoside-binding protein family, has been shown to play a pivotal role in host–pathogen interactions and viral infections [14]. The human Gal3 gene (*LGALS3*) contains six exons and five introns, spanning 17 Kb, with an open reading frame of 750 bp translated into a protein of 250 amino acids with an approximate molecular weight of 30,000 [67]. Of the fifteen galectin members with proto-, tandem-repeat-, and chimera-type structures, Gal3 is the only member of the latter type characterized by three domains: a C-terminal carbohydrate recognition domain (CRD), a highly conserved short N-terminal domain (NTD) with 12 amino acids, and a long NTD rich with proline and glycine [68,69,70,71] (Figure 3).

Gal3 can form dimers and pentamers through its NTD at a high concentration [71]. Gal3 can be found both extracellularly and intracellularly in the cytoplasm, transport vesicles, and nucleus [69,71], and, thus, it is involved in diverse biological activities, including cell growth, pre-mRNA splicing, cell adhesion, cell–cell interactions, apoptosis, angiogenesis, and inflammation [69,71]. Gal3 plays a pivotal role in various stages of viral infections, including viral attachment and entry. Gal3 also mediates inflammatory responses and causes an imbalanced production of pro-inflammatory cytokines [72].

### 3.1. Role of Gal3 in Viral Infection, Including COVID-19

Viral entry is a crucial step in the viral infection process, and it involves the interaction of viral proteins with host cell receptors. In human immunodeficiency virus (HIV), Gal3 is involved in the interaction with the viral envelope glycoprotein gp120, facilitating its binding to host cells through surface receptors, such as CD4, and subsequently promoting infection. It was found that intracellular Gal3 could promote HIV-1 budding through the interaction of ALG-2-interacting protein X (Alix) with viral Gag p6 in T cells [73]. It was also found that Gal3 could regulate virological synapse formation and facilitate intercellular HIV-1 transfer among CD4 T cells, providing an alternative pathway for HIV-1 infection [74]. Additionally, high Gal3-expressing exosomes derived from HIV-1-infected dendritic cells were shown to facilitate HIV-1 infection and dissemination through fibronectin and Gal3-mediated cell fusion [75]. These findings suggest that Gal3 plays a critical role in regulating viral release and synapse formation in HIV-1 infection, and, thus, Gal3 may represent a promising therapeutic target for HIV-1 infection.

In ocular infections, Gal3 was found to be involved in the entry and attachment of herpes simplex virus-1 (HSV-1), as the inhibition of Gal3 impaired HSV-1 infectivity in human corneal keratinocytes [76]. At the onset of influenza A virus (IAV) infection, Gal3—expressed and secreted in airway epithelial cells—bound strongly to the hemagglutinin (HA) protein of IAV and Streptococcus pneumoniae [77]. In addition, the neuraminidase of IAV could desialylate airway epithelial cells and enhance pneumococcal adhesion via galectin interactions. These results suggest that Gal3 could contribute to pneumococcal pneumonia after influenza infection. It was also found that the increased expression of Gal3 could also promote lung inflammation in mice infected with avian H5N1 IAV through the activation of the nucleotide oligomerization domain-like receptor protein 3 (NLRP3) inflammasome [78].

Gal3 plays a vital role in the entry of SARS-CoV-2 into host cells, and, thus, the blocking of Gal3 may prevent the progression of COVID-19 [72]. The S1 subunit of the SARS-CoV-2 spike protein, which is critical for viral entry into host cells, contains a receptor-binding domain (RBD) at the C-terminal domain (CTD). The role of the CTD in viral entry is well established, as it binds to angiotensin-converting enzyme 2 (ACE2) receptors [79]. Caniglia et al. proposed a dual attachment model, which states that the interaction between the NTD of the S1 spike protein (S1-NTD) and host sialic acids may be important for viral entry to host cells as a means of stabilizing the interaction between the S1-CTD and ACE2 [72] (Figure 4). Interestingly, the S1-NTD is structurally similar to the galectin fold [72,80]. The structural similarity of the S1-NTD to human Gal3 has led to the hypothesis that existing Gal3 inhibitors can be used as therapeutic drugs to block NTD–sialic acid interactions [72].

It is worth discussing the interesting correlation between the increased mortality in COVID-19 patients with diabetes (12–22%) and hypertension (23.7–30%) [81,82,83,84] and the increased expression of Gal3 in prediabetic, diabetic, and hypertensive patients in the blood serum, lungs, alveolar cells, and respiratory tract mucus [16,85,86,87,88]. In general, patients with severe COVID-19 show a high concentration of Gal3, which can serve as a potential biomarker for predicting COVID-19 severity, prognosis, and treatment response, as suggested by multiple studies [89,90,91,92]. Furthermore, the spike protein (both S1 and S2 subunits) is a glycoprotein containing one *O*-glycan and multiple *N*-glycan chains [93], which can be readily bound by Gal3 (see Figure 4). In our studies, we confirmed Gal3 binding to the SARS-CoV-2 spike glycoprotein. Thus, it is possible that host Gal3 might establish additional interactions with the virus spike glycoprotein for further stabilization, contributing to prolonged infection and the severity of the disease. These double- and triple-attachment models suggest that the specific inhibition of Gal3 could be a promising therapy for COVID-19.

### 3.2. Immunomodulation of Gal3 in Viral Infection, Including COVID-19

For enveloped viruses such as coronaviruses, there are two critical steps that need to be completed for the successful entry of viruses into host target cells: (1) binding to a host cell receptor and (2) viral envelope fusion with the host cell membrane [94,95,96]. In the latter process, i.e., during the fusion of the viral envelope, the viral genome is released into the cytoplasm, enabling viral replication. Both steps of viral entry to host cells are mediated by a heavily glycosylated class I fusion protein (S protein), which is present in the virus envelope [97].

During viral infections, Gal3 modulates the immune response through the activation and recruitment of various immune cells, such as dendritic cells, macrophages, T cells, and neutrophils [17]. Gal3 can promote the production of pro-inflammatory cytokines and chemokines, such as TNF-α and IL-6, and it can increase neutrophil migration toward infection sites and neutrophil recruitment to the lungs during influenza and Streptococcus pneumoniae co-infection [17].

In the context of SARS-CoV-2 infection, Gal3 is believed to modulate the host immune response, potentially contributing to disease severity [14] through the high-level production and release of pro-inflammatory cytokines, such as TNF-α, IL1β, and IL-6 [66]. Elevated levels of these cytokines are found in patients with severe COVID-19, leading to a “cytokine storm”, ARDS, and multi-organ failure [98,99]. During SARS-CoV-2 infection, Gal3 can activate the NLRP3 inflammasome, which is a critical component of the innate immune response that detects and responds to pathogens [100]. The activation of the NLRP3 inflammasome can induce the production and release of pro-inflammatory cytokines, including IL-1β and IL-18, which can contribute to the overall inflammatory response [92] (Figure 5). Several studies have shown a significant association between high levels of Gal3 in the blood serum and COVID-19 severity [84,85,86], along with increased pro-inflammatory cytokines (IL-1β, TNF-α, and IL-12), chemokine C-C, and chemokine receptor type 5 (CCR5) expression in T cells [92]. Overall, studies suggest that Gal3 may contribute to the acquired pro-inflammatory immune response and intensify the innate pro-inflammatory immune response in severe COVID-19 patients with or without pre-existing conditions or co-morbidities.

Gal3 has been shown to influence the NF-κB signaling pathway, a critical transcription factor known to regulate genes involved in immune and inflammatory responses [101]. Gal3 can activate NF-κB and produce pro-inflammatory cytokines or cytokine storms, as observed in severe cases of COVID-19 [98,102,103]. Gal3 can also modulate other signaling pathways involved in various cellular processes, such as JAK/STAT (Janus kinase/signal transducers and activators of transcription), ERK (extracellular signal-regulated kinase), and AKT (protein kinase B) [104]. Gal3 can dysregulate these pathways, leading to impaired immune responses and contributing to disease pathogenesis during viral infections. We showed the binding of Gal3 to plastic-adsorbed TLR4 (unpublished data). Thus, in viral infections, Gal3 can interact with TLR4 and regulate the NLRP3 inflammasome, thereby promoting inflammasome assembly and activation, which potentially contributes to enhanced inflammation and tissue damage [14,101,105]. Gal3 can also modulate the expression of suppressor of cytokine signaling 1 (SOCS1), which is involved in the overall inflammatory response, and it can influence retinoic acid-inducible gene I (RIG-I) expression during influenza and Streptococcus pneumoniae co-infection, resulting in the dysregulated expression of pro-inflammatory cytokines [77]. Overall, Gal3’s fascinating role in immune signaling pathways underscores its utility as a therapeutic target for COVID-19.

## 4. Development of MyD88 and Gal3 Inhibitors to Treat COVID-19

As discussed above, MyD88 upregulation impairs the host antiviral IFN-β response (type I) [11,25,30,55] (see Figure 2) by interfering with signaling through the interaction of IRF3/IRF7, for example, in the TLR3-TRIF-IRF-dependent pathway. Further, MyD88 induction is known not only for the activation of NF-κB and the subsequent expression of IL-6, TNF-α, IL-1β, etc., but also for the downregulation of type I interferon (double-edged sword) [11,25,55,106,107,108]. In the case of COVID-19, an impaired type I interferon response associated with a persistent blood virus load and an exacerbated inflammatory response driven by NF-κB, particularly by increased TNF-α and IL-6 production, were observed in a cohort of fifty COVID-19 patients [109,110,111]. However, galectin-like S1-NTD interactions with host sialic acids may be critical for SARS-CoV-2 cell entry as a means of stabilizing the interaction between the S1-CTD and ACE2 [72,79,80] (see Figure 4). Moreover, it was found that an increased expression of host Gal3 could promote viral attachment in the airway through interactions with virus spike glycoproteins, and it could promote a pro-inflammatory response. During SARS-CoV-2 infection, an increased expression of Gal3 could activate the NLRP3 inflammasome through TLR4 [100], resulting in the production and release of pro-inflammatory cytokines, which contribute to the overall inflammatory response [92] (see Figure 5). Therefore, we hypothesize that the specific inhibition of Gal3 and MyD88 may represent a promising therapeutic strategy for the treatment of COVID-19.

Given that the BB loop region in the TIR domain is involved in dimerization, which is necessary for MyD88-mediated signaling [10,24,26], many of the MyD88 inhibitors that have been developed are based on the inhibition of either adaptor–adaptor homodimerization or receptor–adaptor heterodimerization. Originally, a synthetic molecule, hydrocinnamoyl-L-valyl-pyrrolidine (AS1), mimicking the BB loop representing consensus RDVLPGT (aa196-202), was shown to disrupt TLR/IL-1R signaling, particularly by disrupting MyD88 and its associated IL-1R [112]. Later, another small molecule (named compound 1) mimicking the BB loop was shown to interfere with MyD88-mediated signaling [113] (Figure 6). Compound 1 was later modified by adding an aromatic benzene ring, and the modified product (named EM-163) was found to be more effective than compound 1 in both cell and animal studies [114]. Further, to increase flexibility in terms of binding to the exposed BB loop of the TIR domain, we synthesized a dimeric compound, named compound 4210. Compound 4210 is composed of two modules of compound 1, which are covalently linked by a non-polar cyclohexane ring [115,116].

Compound 4210 was found to restrict severe pro-inflammatory IL-6, TNF-α, etc., but exhibit type I IFN-inducing properties and demonstrate antiviral activity through the upregulation of IFN-β and RANTES in various viral infections and mouse models of diseases [25]. Overall, studies have shown that compound 4210 can function as an immune-modulating molecule through the deactivation of MyD88, adjust biological pathways, and accelerate type I—and potentially type III—IFN signaling through phosphorylated IRFs for a strong antiviral IFN response [25] (see Figure 2). Similar to compound 4210, a few other small-molecule inhibitors of MyD88 (such as T6167923 and S5, capable of blocking the TIR domain homodimerization of MyD88) [21,66] have also shown type I-inducing properties (unpublished data). Overall, these data suggest that, during viral infection, targeting MyD88 inhibition may facilitate an increased IFN response.

Several Gal3 inhibitors and antagonists, for example, small-molecule carbohydrates such as TD139 (GB0139) and GB1211 (from Galecto, galecto.com), large-molecule natural products such as GR-MD-02 and GM-CT-01 (from Galectin Therapeutics, galectintherapeutics.com), Prolectin-series drugs (BioXyTran, bioxytraninc.com), and GM100-series biologics (from GlycoMantra, glycomantra.com), are being developed for therapeutic applications in various diseases [71] (see Figure 6). With reference to COVID-19, TD139 and GB0139 (DEFINE, NCT04473053) are being investigated in a Phase 2 trial to evaluate their efficacy in pre-ventilator COVID-19 patients (n = 200). These are dosed with 5 mg twice daily for the first two days and then 5 mg once daily for the remaining twelve days or are examined until discharge or withdrawal from the hospital or trial. This study aims to determine whether treatment with inhaled GB0139 can reduce viral load and disease severity, along with any changes in blood biomarkers. Although the trial is ongoing, the preliminary results from about one-quarter of the COVID-19 patients (n = 40) treated with GB0139 plus SOC (standard of care) are encouraging [118]. This drug combination (GB0139 + SOC) was well tolerated and achieved clinically relevant endpoints (a reduction in markers associated with inflammation, coagulation, fibrosis, etc., compared with the SOC alone), underscoring the efficacy of GB0139 in COVID-19 patients [118]. In a double-blind, placebo-controlled clinical study (NCT04512027), ProLectin-M (a galactomannan-based oral drug developed from guar gum by BioOxyTran) was performed in a small cohort of COVID-19 patients (n = 10) with mild-to-moderate disease symptoms for a week to determine whether it caused a reduction in the viral copy numbers. The results were encouraging, as the patients in the treatment group, but not the placebo group, were found to be RT-PCR-negative for SARS-CoV-2 from day 3 onwards [119]. We developed recombinant glycoproteins that target Gal3 with picomolar affinity (GM100 series, patent pending) based on a few modifications of our patented drug TFD100 [71,120]. The interactions of Gal3 with its endogenous ligands are typically strong (in the nanomolar range) [71], and, thus, our Gal3 antagonists can out-compete Gal3’s intrinsic interaction with its endogenous ligands. We demonstrated the pre-clinical efficacy of our Gal3 antagonists in various indications, such as metastatic prostate cancer, liver fibrosis, and type 2 diabetes (glycomantra.com), in relevant animal models without any adverse side effects. Our GM100-series drugs could be used alone or in combination in various Gal3-mediated diseases. 

## 5. Concluding Remarks

Overall, innate immunity constitutes the first line of defense against invading pathogens, maintaining normal microbiota and homeostasis. The discovery of host-directed therapeutic products can be beneficial in balancing the host’s internal defense mechanisms against disease, limiting excess inflammation, or both, resulting in improved disease outcomes such as fewer infection-associated deaths. Here, we discuss the possible strategies for discovering small-molecule antiviral therapeutics for COVID-19 and potentially emerging coronaviruses. A MyD88-targeted therapeutic approach in a Phase 1 clinical trial against COPD has been proven to be successful in controlling inflammatory diseases, and this has been validated [121]. Our studies revealed the mechanisms underlying the impairment of antiviral type I IFN signaling and the role of upregulated MyD88 (MyD88–IRF interactions), which likely influence alternative MyD88-independent immune signaling pathways during many viral infections [25]. The therapeutic inhibition of MyD88 has been shown to restore the host antiviral type I IFN response in MyD88-independent pathways [11,25,55]. Data suggest that host-directed therapy can feasibly modulate the immune response by stimulating host defense mechanisms and targeting pathways that are perturbed because of pathogen exposure. Therefore, targeting host factors such as MyD88 involved in the interference of innate immune modulation such as type I and III IFNs offers the opportunity for discovering broad-spectrum antiviral drugs. Here, we discuss the notion that pharmacological blockade with short-term treatment with MyD88 inhibitors could induce type I IFN while reducing the acute exacerbation of inflammatory cytokines. Various diseases, including toxic shock, COPD, and COVID-19, are more like manifestations of an imbalance in host innate immunity without the control of inflammation. In this context, MyD88-targeted therapy would likely be advantageous for controlling severe inflammatory responses. The available data suggest the benefit of MyD88-targeted therapy with broad-spectrum antiviral type I IFN-inducing properties. However, more focused and in-depth studies are needed to determine the suitability of this approach in limiting inflammation-associated syndromes and infections in complex diseases such as COPD and COVID-19.

In addition, Gal3 has emerged as a multifaceted player in host–pathogen interactions and viral infections. Gal3 functions not only as a PRR but also as a DAMP. In the context of SARS-CoV-2 infection, Gal3 has been shown to modulate the host immune response, potentially contributing to the severity of COVID-19. Gal3 can activate the NLRP3 inflammasome, a critical component of the innate immune response that detects and responds to pathogens. The activation of the NLRP3 inflammasome by Gal3 results in the production and release of pro-inflammatory cytokines, such as IL-1β, IL-18, and IL-33, contributing to the overall inflammatory response during SARS-CoV-2 infection. The therapeutic potential of Gal3 inhibitors, namely, GB0139 and Prolectin-M, in hospitalized patients with COVID-19 has already been established. As a result, strategies ranging from the design and preparation of potent synthetic small-molecule antagonists (i.e., glycomimetics) to the acquisition of large biologics from natural sources are being employed to target Gal3 for therapeutic intervention against viral diseases, as well as other inflammatory diseases. These Gal3 inhibitors and MyD88 inhibitors can be used separately as standalone drugs or in combination to obtain optimal results against viral infections, including SARS-CoV-2. 

## Figures and Tables

**Figure 1 ijms-25-08421-f001:**
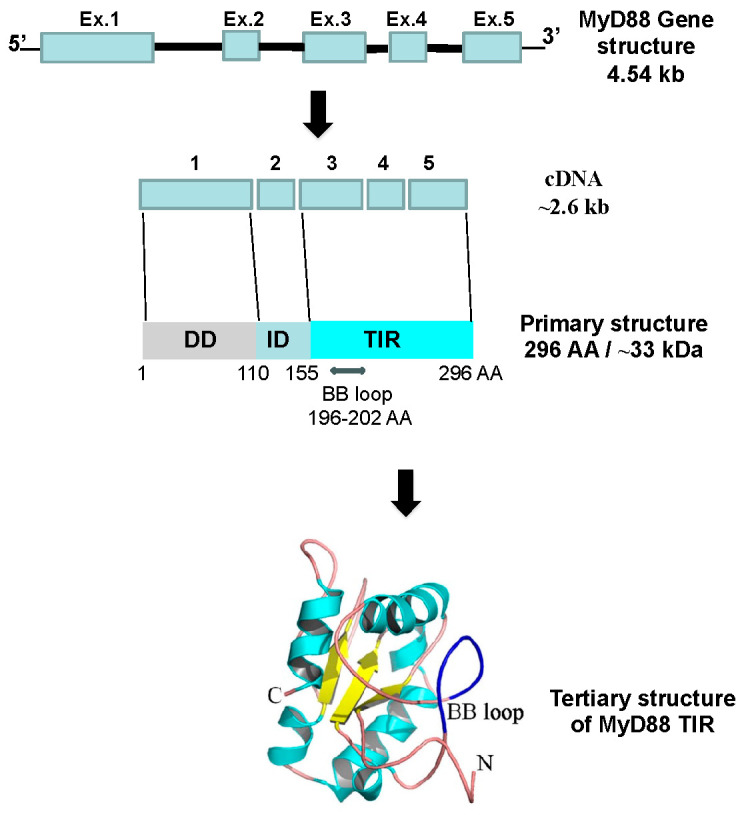
Schematic representation of MyD88 primary and tertiary structures. The 3D model of the MyD88 TIR domain and the BB loop region (in dark blue) was adapted from Refs. [12,21]. DD, death domain; ID, intermediary domain; TIR, Toll-interleukin-1 receptor.

**Figure 2 ijms-25-08421-f002:**
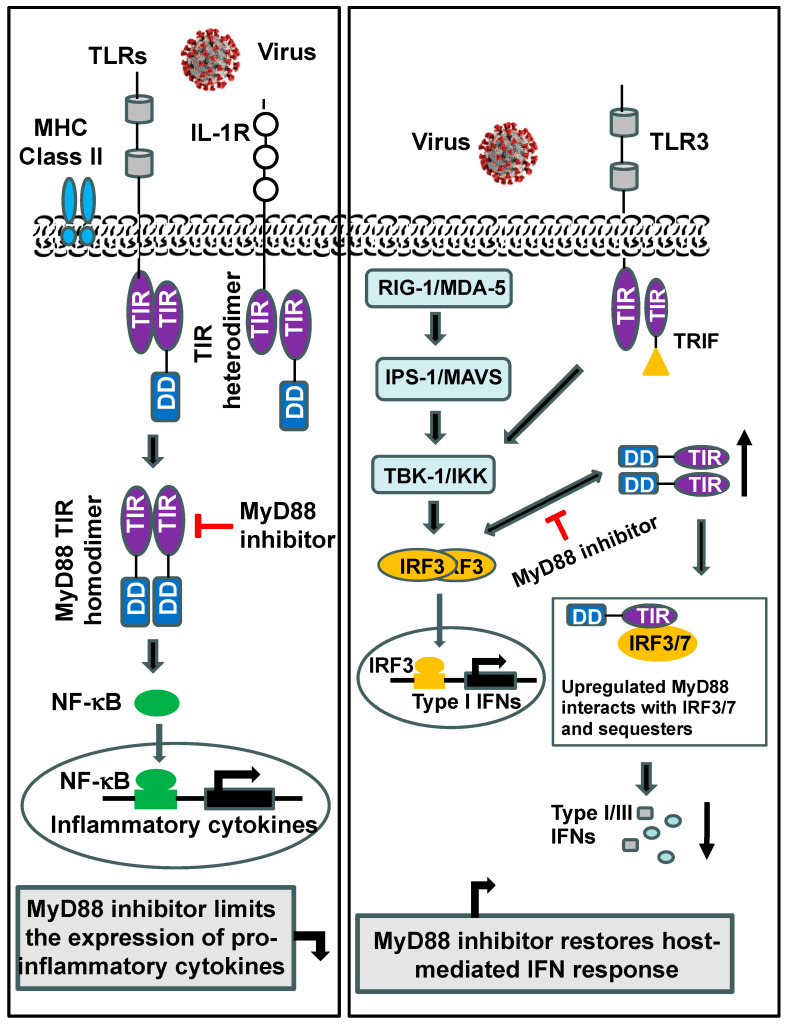
Schematic representation showing MyD88-mediated pro-inflammatory response and plausible mechanism of MyD88 inhibition in restoring host-mediated immune responses.

**Figure 3 ijms-25-08421-f003:**
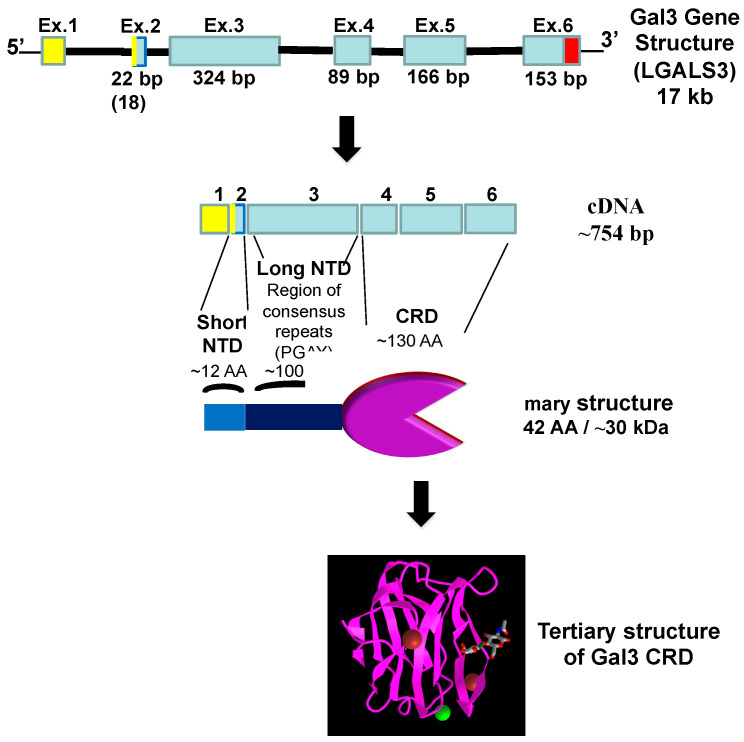
Schematic representation showing primary and tertiary structures of Gal3. The tertiary structure of Gal3 CRD bound to LacNAc (N-acetyllactosamine) was obtained from the NCBI (https://www.ncbi.nlm.nih.gov/Structure/pdb/1KJL, access on 21 May 2024).

**Figure 4 ijms-25-08421-f004:**
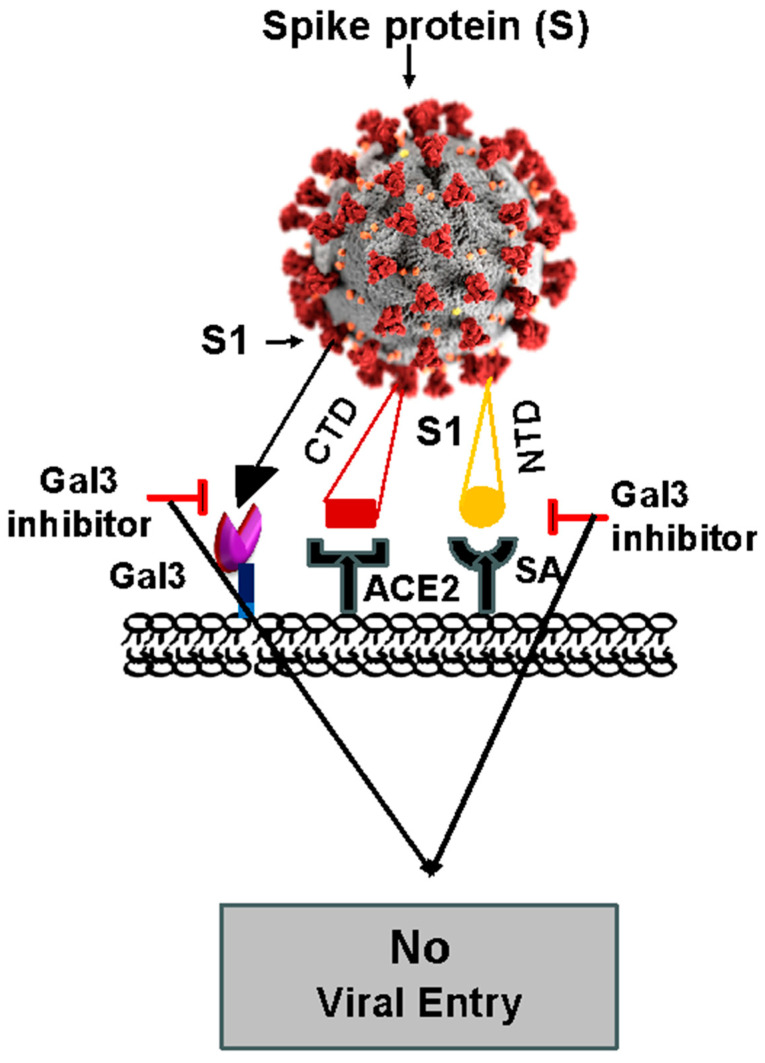
Schematic representation showing virus–host cell interactions and Gal3 inhibition blocking viral entry to host cells. Gal3 is believed to play critical role in viral entry to host cells. Interaction between the galectin-like S1-NTD (N-terminal domain of the virus S1 spike protein) and host sialic acids could be critical for viral entry as means of stabilizing the interaction between S1-CTD (C-terminal domain of the virus S1 spike protein) and ACE2. Moreover, host Gal3 may participate in additional interactions with virus spike glycoprotein for further stabilization, contributing to prolonged infection and severity of disease. Therefore, Gal3 inhibition may disrupt attachment of SARS-CoV-2 to cell surface, preventing entry to host cells (adapted from Refs. [72,80]). The SARS-CoV-2 image was taken from the Centers for Disease Control and Prevention (CDC) website (https://stacks.cdc.gov/view/cdc/86942, access on 20 May 2024).

**Figure 5 ijms-25-08421-f005:**
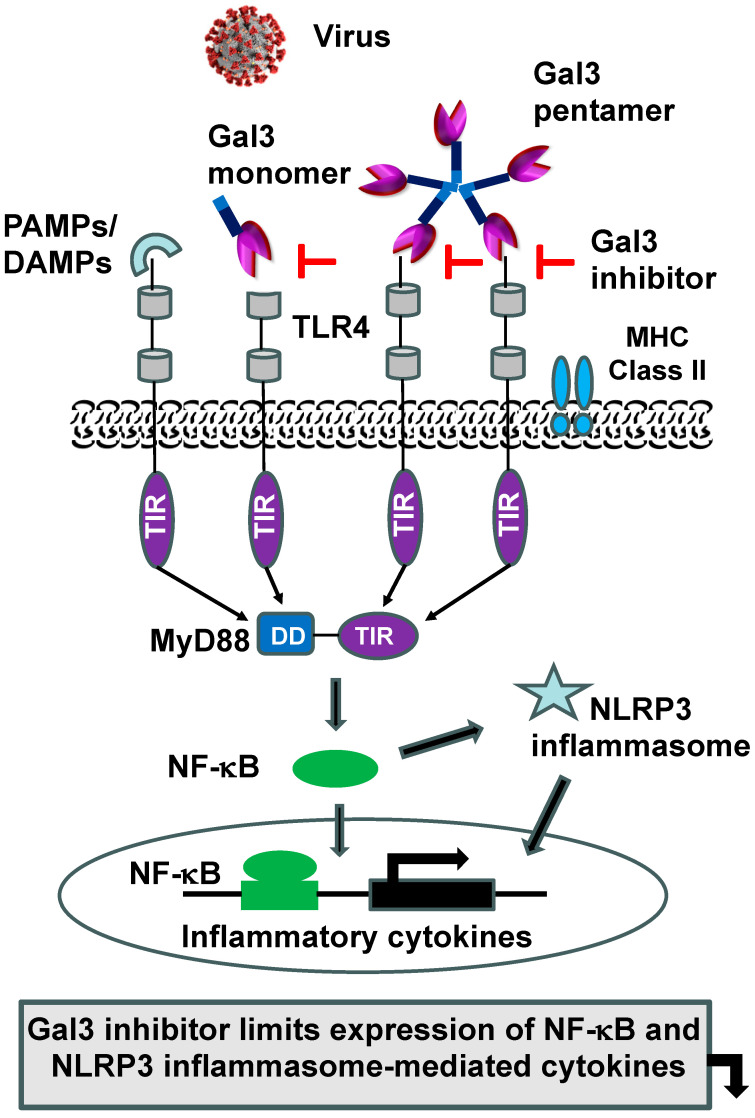
Schematic representation of Gal3-mediated upregulation of pro-inflammatory cytokines and Gal3 inhibition reversing it. During viral infection, Gal3 can interact with TLR4 to elicit NF-κB-mediated pro-inflammatory cytokines. Gal3 can also activate NLRP3 inflammasome, leading to secretion of pro-inflammatory cytokines. However, inhibition of Gal3 can prevent activation of both NF-κB and NLRP3 inflammasome, abrogating pro-inflammatory signaling (according to Ref. [14]). SARS-CoV-2 image was taken from the Centers for Disease Control and Prevention (CDC) (https://stacks.cdc.gov/view/cdc/86942, access one 20 May 2024).

**Figure 6 ijms-25-08421-f006:**
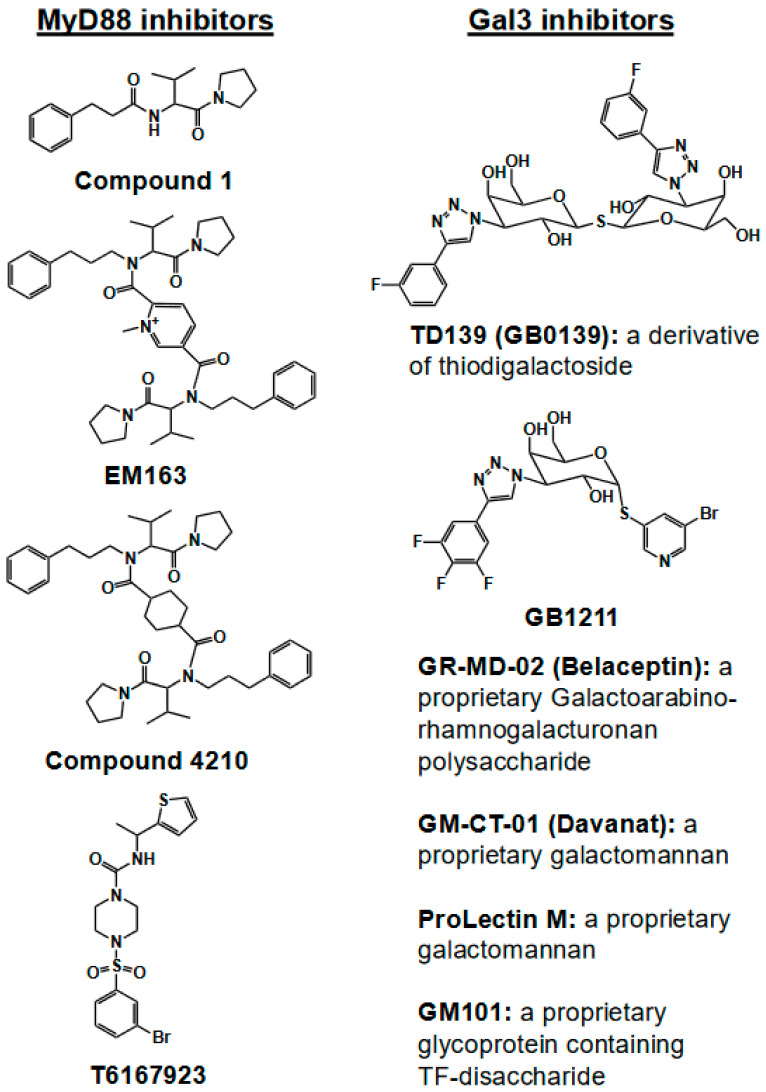
Chemical structures and compositions of MyD88 and Gal3 inhibitors. Structures of compound 1, EM163, and compound 4210 were adapted from Ref. [11]. Structure of T6167923 was adapted from Ref. [21]. Structures of TD139 and GB1211 were adapted from Ref. [117].

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
