# Peer review of "Targeting Myeloid Differentiation Primary Response Protein 88 (MyD88) and Galectin-3 to Develop Broad-Spectrum Host-Mediated Therapeutics against SARS-CoV-2"

_ijms, 2024, doi:10.3390/ijms25158421_

Round 1

Reviewer 1 Report

Comments and Suggestions for Authors

This is a review paper arguing that the increased expression of MyD88 induced by viruses is responsible for the reduced production of the anti-viral Type I and TypeIII interferons. They also argue that Galectin-3 is involved in the activation of TLRs resulting in the cytokine storm seen in SARS-CoV-2 infected persons. Extensive English editing is required. The text is in general quite narrative and superficial without going into depth. There is a lack of data, requiring in-depth description of all issues mentioned. There are jumps from subject to subject, which are intermingled. The manuscript should be more focused and organized.

No need for running title.

Many clichés. Double wording such as "dysregulated uncontrolled";  "restoring and balancing".

Line 45: no need for "etc" when writing "such as".

Line 45: Correct to IL-1β.

Line 47-48 raise an issue that the authors wrote "is not adequately addressed". Then I would expect in the next paragraph to get information about it, but instead a general quite non-informative sentence appears.

Line 57: Correct to "initiates"

Line 65: In addition to Ref 10, the following reference should be added: doi: 10.12703/P6-97

Line 66: Starts with the wordings: "On the other hand", but it is not in contrast to TLR.

Line 78: Provide definition of "TIR".

Figure 1 can be made more informative: The 5 exons can be depicted with the 4 introns. And show which exon is responsible for which part of the protein. It is querying to show not only the 3D structure of the TIR domain, but also the 3D structure of MyD88. And mention what are the features of the TIR domain. All MyD88 interacting partners should be shown after Figure 1.

Figure legend 1: Should include the different abbreviations in the figure.

The text after Figure 1 speaks about other domains not shown in Figure 1. Please add these to the figures (e.g., INT domain, BB-loop etc).

Line 85: It says that: " The C-terminal TIR domain binds to the receptor TIR".  Please check what you mean by receptor here. A reference should be added to the sentence (e.g., doi: 10.2174/138920312804871148).

Figure 2 should be presented after the description in the text.

Figure 2 should first describe how virus activates the TLR signaling pathways and add all signaling pathways activated, and not only the NFκB pathway.

Figure 2 is incomplete, and all names of mediators should be described in the legend (e.g., RIG-1, MAVS, TBK-1).  All components of Figure 2 should be described in the text.

Whitin the figure it says that "upregulated MyD88 interacts with" "Upregulated" an not "interact" MyD88 interacts. (The difference in resting and activated MyD88 should be stated). Does MyD88 in resting state also interact with IRFs? Is this a matter of amount of MyD88 or different activation? How does the homodimerization of MyD88 occur? Does the monomer or the dimer of MyD88 interact with IRFs? How does the MyD88 inhibitor prevent the interaction between MyD88 and IRFs? All these need to be described here. The mechanism of MyD88 upregulation should be described. (e.g., https://doi.org/10.3390/v16040601). Figure 2 should also include IRAK and TIRAP.

Line 110: The dysregulation of MyD88 should be described, especially by viral infections.

Line 116, "but" can be deleted since you say "while" in the beginning of the sentence.

Line 136: Do you mean "recognition of virus"? or you indeed mean recognition of "replication"?

Line 139-and further – this should be more precise.  It is not clear "are like".  Each of the signaling should be described in depth.

Line 147- and further should appear previously under the description of viral infection activation. Then describe the different pathways activated, and only after then to describe strategies to dampen the excessive immune responses.

A figure should be made to summarize the responses to viral infections, and how IFN Type I and Type III are acting. This is partly described a little bit later (lines 190 and further). There is some repetions, and these sections should be put together. And illustrated in a figure.

Line 174: After Hatten et al. the reference should be added.

Line 178: Describe here which modulations were done.

What are the adverse effects of IFN therapy?

Line 185: k should be kappa.

Line 223: Define VEEV.

Lines 226-227: It is not clear: " MyD88 interaction with IRF3/IRF7 in a MyD88-independent immune signaling pathways" How can MyD88 act in a MyD88-independent manner? The further sentence is not clear. Previously you said that increased MyD88 causes excessive cytokine expression, and here you say, "linked to weak immune signaling". You need to better specify what you want to say.

There are many backs and forth, and not always clear what the authors want to say. I would suggest the authors ponder on what do they want to state, and then organize the text accordingly. In the present form it is jumping from one issue to the other issue.

I would suggest collecting all sections with MyD88 inhibition into one section.

Line 248-and further should go together with a section of anti-viral responses.

Sentence in lines 251-further is a repetition. Also, systems cannot be molecules.

Line 255: what do you mean by "mediating resistance viruses"? Some words are lacking here.

Coming to the end of Section 2.1, the title does not conform with the text. The Title is nice, and the reader is looking for the comparison which should be better emphasized.

Line 265 is again a repetition of previous wordings. The section should be focused on the content of the title. This text should appear together with a description of SARS-CoV pathogenesis in a section for itself. (Should be one of the earlier sections).

Section 2.2 has many repetitions of the previous text. Please put each topic into each section, to avoid repetition.

The whole section from 262-302 is repetition and does not discuss what the title indicates.

The text 262-302 should be fused with sections describing these issues.

Compound 4210 is first described in line 97, then MyD88 inhibition is mentioned in line122, and then in Section 2.2 in lines 306 and 319. The structure of the inhibitors should be shown. Have these inhibitors been tested in a SARS-CoV model? This is actually what the readers think from the title of the manuscript.

Figure 3: Same as for figure 1, also here are a correlation between exons and protein domains should be provided as well as the 3D of the whole Galectin-3 molecule. The interacting partners of Galectin-3 should be illustrated.

Figure 5: A spelling mistake: correct to monomer.

Reference should be added to line 345.

In line 464 you specifically state the beta form, while previously in the text you emphasize Type I and Type III. Is there any reason for it.

Section 4 starts with an initial description which has been discussed in the previous section. Quite a repetition.

Line 479: Why is there a strikethrough of Figure 5?

Line 482: Spelling mistake: correct to "the" (written "he").

The formula of the different MyD88 and Galectin inhibitors should be shown.

Line 484: I would suggest writing: "many" instead of "lot".

According to the manuscript – MyD88 has both beneficial and inhibitory activities during viral infection. Do the MyD88 inhibitor only affect the activation of IFRs? Or do they also inhibit the beneficial MyD88-signaling pathway?

Line 593: correct to beta.

Ine 611: correct to kappa.

Line 616: There is no limited space in this journal. Many references are lacking, and more in-depth discussion should be done.

Some references are underlined. These underlines should be removed.

Comments on the Quality of English Language

Moderate English editing required.

Author Response

Reviewer #1:

Reviewer 1 comments:

Comments and Suggestions for Authors

This is a review paper arguing that the increased expression of MyD88 induced by viruses is responsible for the reduced production of the anti-viral Type I and Type III interferons. They also argue that Galectin-3 is involved in the activation of TLRs resulting in the cytokine storm seen in SARS-CoV-2 infected persons. Extensive English editing is required. The text is in general quite narrative and superficial without going into depth. There is a lack of data, requiring in-depth description of all issues mentioned. There are jumps from subject to subject, which are intermingled. The manuscript should be more focused and organized.

Response: The manuscript is an interpretive synthesis review article addressed to explore the potential of developing therapeutic strategies against viral infections for restoring host-mediated immunity as a standalone strategy or in combination with other therapeutic modalities targeting viral components. This type of host mediated therapeutic approach could be used where infections such as SARS-CoV-2 or other viral infection causes a dys-regulated immune response in host that lead to disease pathology such as COVID-19. The authors agree with the reviewer that this review is a narrative i.e. an account of connected events focused on host factors in the context of COVID-19 where host-mediated unbalanced immune signaling pathways were severely perturbed or interfered by the host factor(s) of other signaling pathways. Over the years a vast amount of scientific literature is available that give in depth descriptions of the host components involved in inducing host immunity including MyD88 and Galectin-3.  We also cited many of these references in this review.  Given the importance of these two host factors in virus entry, innate immune signaling and balancing host immunity, and availability of recently developed inhibitors, our review is focused on these two host factors in exploring applicability and prospect of discovering antiviral therapeutics. We provided a significant description of these host targets, it’s’ role and regulation in host innate immune signaling pathways in the context of several viral infections, Therefore, we believe the reviewer will agree, that this is an interpretive synthesis review in which we outlined a possible therapeutic platform which may potentially offers to treat a broad range of viral diseases including COVID-19.

  1. No need for running title.

Response: As suggested by reviewer, we removed the running title.

  1. Many clichés. Double wording such as "dysregulated uncontrolled";  "restoring and balancing".

Response: We corrected the sentence of double wording by placing a more appropriate wording

  1. Line 45: no need for "etc" when writing "such as".

Response: Corrected

  1. Line 45: Correct to IL-1β.

Response: Corrected

  1. Line 47-48 raise an issue that the authors wrote "is not adequately addressed". Then I would expect in the next paragraph to get information about it, but instead, a general, quite non-informative sentence appears.

Response: In the revised manuscript, this his has been updated with references 5 and 6.

  1. Line 57: Correct to "initiates"

Response: Corrected

  1. Line 65: In addition to Ref 10, the following reference should be added: doi: 10.12703/P6-97

Response: As suggested by the reviewer, now we added the new reference (Ref. 13 in the revised manuscript).

  1. Line 66: Starts with the wordings: "On the other hand", but it is not in contrast to TLR.

Response: We agree with the reviewer, this is a new paragraph and not in contrast to TLR. We, therefore, removed “on the other hand” and corrected the sentence which begins with “In addition,-----------“.

  1. Line 78: Provide definition of "TIR".

Response: We described the TIR as Toll-interleukin-1 receptor domain in the revised manuscript.

  1. Figure 1 can be made more informative: The 5 exons can be depicted with the 4 introns. And show which exon is responsible for which part of the protein. It is querying to show not only the 3D structure of the TIR domain, but also the 3D structure of MyD88. And mention what are the features of the TIR domain. All MyD88 interacting partners should be shown after Figure 1.

Response: Now we modified the Fig. 1, indicated BB loop region and described in the figure legend. We presented 3D structure of the TIR domain only as it interacts with the MyD88 inhibitors such as 4210 through its BB-loop.

  1. Figure legend 1: Should include the different abbreviations in the figure.

Response: Now we described all abbreviations in the legend.

  1. The text after Figure 1 speaks about other domains not shown in Figure 1. Please add these to the figures (e.g., INT domain, BB-loop etc).

Response: In the text after Fig.1, we mentioned the BB loop region in the TIR domain with its amino acid sequence.

  1. Line 85: It says that: " The C-terminal TIR domain binds to the receptor TIR".  Please check what you mean by receptor here. A reference should be added to the sentence (e.g., doi: 2174/138920312804871148).

Response: We clarified this sentence.  We also added a new reference (No. 22) as suggested.

  1. Figure 2 should be presented after the description in the text.

Response: The Fig. 2 is now inserted after it is mentioned in the text.

  1. Figure 2 should first describe how virus activates the TLR signaling pathways and add all signaling pathways activated, and not only the NFκB pathway.

Response: In the revised manuscript under Fig. 2, we added a brief description of how viruses activate immune signaling pathways including TLRs.

  1. Figure 2 is incomplete, and all names of mediators should be described in the legend (e.g., RIG-1, MAVS, TBK-1).  All components of Figure 2 should be described in the text.

Whitin the figure it says that "upregulated MyD88 interacts with" "Upregulated" an not "interact" MyD88 interacts. (The difference in resting and activated MyD88 should be stated). Does MyD88 in resting state also interact with IRFs? Is this a matter of amount of MyD88 or different activation? How does the homodimerization of MyD88 occur? Does the monomer or the dimer of MyD88 interact with IRFs? How does the MyD88 inhibitor prevent the interaction between MyD88 and IRFs? All these need to be described here. The mechanism of MyD88 upregulation should be described. (e.g., https://doi.org/10.3390/v16040601). Figure 2 should also include IRAK and TIRAP.

Response: As suggested by the reviewer, we described all components in Fig. 2 legend. We corrected the sentence as suggested by the reviewer. Previous reports from our laboratory suggested that upon stimulation with poly I:C (dsRNA), MyD88 was upregulated (amount) and this cellular up regulation caused IRF3/IRF7 interaction but in the presence of MyD88 inhibitor (compound 4210) increased phosphorylation IRF3/IRF7 (for details, please see Refs 25, 55). Other investigators also described that viral infection caused several fold increase in MyD88 [28, 30].

Neither the mechanism of MyD88 (mRNA) upregulation nor the reference (https://doi.org/10.3390/v16040601) suggested by the reviewer is not our focus in this review. Here we are focusing the host immune response in the context of viral infection, particularly COVID-19 [59].

  1. Line 110: The dysregulation of MyD88 should be described, especially by viral infections.

Response: We now described the dysregulation of MyD88 by viral infections in detail with specific citations of references.

  1. Line 116, "but" can be deleted since you say "while" in the beginning of the sentence.

Response: Now the sentence is corrected.

  1. Line 136: Do you mean "recognition of virus"? or you indeed mean recognition of "replication"?

Response: It is meant recognition of viral replication.

  1. Line 139-and further – this should be more precise.  It is not clear "are like".  Each of the signaling should be described in depth.

Response: Now we inserted details of IFN response pathways as suggested by the reviewer.

“Briefly, in infected cells, a signaling chain is activated upon detection of viral double stranded RNA (dsRNA) binding to RIG-I or MDA5 which are ubiquitously expressed in most tissues and seem to function in parallel having a degree of virus specificity. RIG-I or MDA5 mediated signaling for interferon response occurs after binding to downstream factors called interferon beta promoter stimulator 1 (IPS-1) or MAVS, which leads to activation of IRF3 kinases, such as TBK-1 or IKKe which are known to phosphorylate IRF-3 or IRF-7.  Phosphorylated IRFs homo-dimerizes and moves into nucleus where its recruits the transcriptional co-activators p300 and CREB-binding protein (CBP) to initiate IFN-b mRNA synthesis. NF-kB and AP-1 are also recruited in dsRNA-dependent way. Ligand-induced triggering of TLR3 by dsRNA proceeds via adaptor molecule TRIF which bypasses IPS-1/MVS and directly activates the kinase TBK-1 and subsequently IFN response as described earlier. Thus, together these transcriptional factors strongly trigger “first wave” IFN response.

  1. Line 147- and further should appear previously under the description of viral infection activation. Then describe the different pathways activated, and only after then to describe strategies to dampen the excessive immune responses.

Response: Updated as suggested by the reviewer.

  1. A figure should be made to summarize the responses to viral infections, and how IFN Type I and Type III are acting. This is partly described a little bit later (lines 190 and further). There is some repetions, and these sections should be put together. And illustrated in a figure.

Response: We already described it.

  1. Line 174: After Hatten et al. the reference should be added.

Response: We inserted a reference.

  1. Line 178: Describe here which modulations were done.

Response: We mentioned the modulator in the revised manuscript a mutated form of IFNalpha 2, termed IFNmod treatment which can elicit weak IFN-I signaling.

  1. What are the adverse effects of IFN therapy?

Response: The question to the authors is unclear about the adverse effect while the IFN therapy is recognized.

  1. Line 185: k should be kappa.

Response: Corrected

  1. Line 223: Define VEEV.

Response: Now it is defined in the revised manuscript “Venezuelan equine encephalitis virus”

  1. Lines 226-227: It is not clear: "MyD88 interaction with IRF3/IRF7 in a MyD88-independent immune signaling pathways" How can MyD88 act in a MyD88-independent manner? The further sentence is not clear. Previously you said that increased MyD88 causes excessive cytokine expression, and here you say, "linked to weak immune signaling". You need to better specify what you want to say.

Response: We provided the following the clarification:

As single infectious agent may harbor multiple PRR agonists, which can trigger different sets of PRRs such as multiple TLRs, RIG-I or MDA 5 initiated innate immune signaling such as TLR3-TRIF-IRF3/IRF7 or RIG-I/MDA5-MAVS axis (MyD88-independent signaling) for antiviral immune responses. Upregulation of MyD88 and interaction with IRF3 or IRF7 limit the availability of IRFs (sequester), leading to less IRF phosphorylation, dimerization and translocation to the nucleus for weak signaling to induce less interferon synthesis. In next paragraph it has been explained in detail.

  1. There are many backs and forth, and not always clear what the authors want to say. I would suggest the authors ponder on what do they want to state, and then organize the text accordingly. In the present form it is jumping from one issue to the other issue.

Response: Now we made it clear through explanation.

  1. I would suggest collecting all sections with MyD88 inhibition into one section. Line 248-and further should go together with a section of anti-viral responses.

Response: Here we discussed about signaling pathways and how signaling interference could happen as a consequence of activation of TLRs initiated events.

  1. Sentence in lines 251-further is a repetition. Also, systems cannot be molecules.

Response: We corrected the sentence.

  1. Line 255: what do you mean by "mediating resistance viruses"? Some words are lacking here.

Response: It meant that when cells are exposed to type 1, IFNs display pronounced resistance to virus replication. This has been corrected in the revised manuscript.

  1. Coming to the end of Section 2.1, the title does not conform with the text. The Title is nice, and the reader is looking for the comparison which should be better emphasized.

Response: The authors agree with the reviewer.  In the last paragraph of section 2.1, we rearranged the text and provided a meaningful explanation that confirms the text.

  1. Line 265 is again a repetition of previous wordings. The section should be focused on the content of the title. This text should appear together with a description of SARS-CoV pathogenesis in a section for itself. (Should be one of the earlier sections).

Response: This is discussed in the context of COVID-19.

  1. Section 2.2 has many repetitions of the previous text. Please put each topic into each section, to avoid repetition.

Response: We modified the text in the revised manuscript.

  1. The whole section from 262-302 is repetition and does not discuss what the title indicates.

The text 262-302 should be fused with sections describing these issues.

Response: In this section, we briefly described the importance of the Type I IFN role in severe pro-inflammatory response with a lack or delayed type 1 response in the context of COVID-19 disease.

  1. Compound 4210 is first described in line 97, then MyD88 inhibition is mentioned in line122, and then in Section 2.2 in lines 306 and 319. The structure of the inhibitors should be shown. Have these inhibitors been tested in a SARS-CoV model? This is actually what the readers think from the title of the manuscript.

Response: In the revised manuscript, we described and provided the structure of the compound 4210. This compound has not been tested yet in COVID-19 model and this was now mentioned in the revised manuscript (Section 2.2 last line of the last paragraph). Data from our laboratory indicated type I inducing properties in cell-based infection assay and mouse model with several viral infections [25, 55].

  1. Figure 3: Same as for figure 1, also here are a correlation between exons and protein domains should be provided as well as the 3D of the whole Galectin-3 molecule. The interacting partners of Galectin-3 should be illustrated.

Response: We now updated Fig. 3.

  1. Figure 5: A spelling mistake: correct to monomer.

Response: Corrected

  1. Reference should be added to line 345.

Response: We added a reference.

  1. In line 464 you specifically state the beta form, while previously in the text you emphasize Type I and Type III. Is there any reason for it.

Response: We used IFN-b because of the specific references cited here, were IFN-b (which is also type I IFN) was measured. We updated this sentence by adding ‘Type I’ after IFN-b.

  1. Section 4 starts with an initial description which has been discussed in the previous section. Quite a repetition.

Response: In the revised version, we cut short the description, which is only mentioned in the context of COVID-19.

  1. Line 479: Why is there a strikethrough of Figure 5?

Response: It was a mistake. Corrected.

  1. Line 482: Spelling mistake: correct to "the" (written "he").

Response: Corrected

  1. The formula of the different MyD88 and Galectin inhibitors should be shown.

Response: Structures of various MyD88 and Gal3 inhibitors (Fig. 6) has been added in the revised manuscript.

  1. Line 484: I would suggest writing: "many" instead of "lot".

Response: The authors used the word “many” instead of” lot”.

  1. According to the manuscript – MyD88 has both beneficial and inhibitory activities during viral infection. Do the MyD88 inhibitor only affect the activation of IFRs? Or do they also inhibit the beneficial MyD88-signaling pathway?

Response: MyD88 inhibitor not only affects the activation of IRFs, but is also beneficial during hyperactivation of immune signaling situations such as septic shock. Reports from our laboratory established that during exposure to staphylococcal enterotoxin SEA or SEB, MyD88 inhibitor provided protection by reducing pro-inflammatory cytokine responses from toxic shock induced death in a mouse model of diseases (For details please see Refs 11, 116]. In a beneficial response i.e. normal immune response pathways MyD88 inhibition is not needed. However, over activation or during dysregulated host imbalance immune responses, it has potential role in restoring the host immune response.

  1. Line 593: correct to beta.

Response: Corrected

  1. Ine 611: correct to kappa.

Response: Corrected

  1. Line 616: There is no limited space in this journal. Many references are lacking, and more in-depth discussion should be done.

Response: Seven additional relevant references [13, 18, 19, 22, 28, 29, 117] were added in the revised manuscript. We also improved the discussion.

  1. Some references are underlined. These underlines should be removed.

Response: Corrected.

Reviewer 2 Report

Comments and Suggestions for Authors

The manuscript is a narrative review, and that makes it difficult to review properly. The topics and themes are well presented, but there are no substantial nor systematic efforts to show the possible problems of downsides. There is no guarantee that the authors did not select only the studies that contributed to their ideas, and therefore, the scope of the manuscript is at risk of being biased. There is no way around this in a narrative review, and that is the reason why systematic review would have been a better option. If we ignore this, the manuscript provides a wide and informative overview of the topic, and provides a possible suggestion on the future studies (notably, the story of Alzhemier and faulty tau protein is a sore reminder how bad narrative review can lead to extraodrinary amount of money being spent on the wrong moelcular target). There is a strikethrough on Figure 5, and it is unclear what the authors meant by this. Some linguistic parts could be improved, for example did not kill butr caused the deaths of. There are similar problems elsewhere. NOTE: there is an unsatisfactory plagiarism share of the principal authors paper (https://doi.org/10.1007/s12026-021-09188-2) that should be corrected. 

Author Response

Reviewer #2:

 The manuscript is a narrative review, and that makes it difficult to review properly. The topics and themes are well presented, but there are no substantial nor systematic efforts to show the possible problems of downsides. There is no guarantee that the authors did not select only the studies that contributed to their ideas, and therefore, the scope of the manuscript is at risk of being biased. There is no way around this in a narrative review, and that is the reason why systematic review would have been a better option. If we ignore this, the manuscript provides a wide and informative overview of the topic, and provides a possible suggestion on the future studies (notably, the story of Alzhemier and faulty tau protein is a sore reminder how bad narrative review can lead to extraodrinary amount of money being spent on the wrong moelcular target). There is a strikethrough on Figure 5, and it is unclear what the authors meant by this. Some linguistic parts could be improved, for example did not kill butr caused the deaths of. There are similar problems elsewhere. NOTE: there is an unsatisfactory plagiarism share of the principal authors paper (https://doi.org/10.1007/s12026-021-09188-2) that should be corrected. 

Response: We appreciate the reviewer comment on the presentation of topics and the importance of the themes. The authors agree with the reviewer that this review is a narrative, an interpretive synthesis review i.e. an account of connected events focused on host factors in the context of COVID-19 where host-mediated unbalanced immune signaling pathways were severely perturbed or interfered by the host factor(s) of other signaling pathways. In regard to the biasness of the idea or validity of the target, a preclinical study in COPD, the validity of the target has been tested with promising results.

Strikethrough on Figure 5 was a mistake. We now corrected.

As suggested by the reviewer we put effort in improving the linguistic parts and avoided unsatisfactory plagiarism by changing the words and rewriting the sentences.

Moderate English editing required.

Response: We edited the manuscript where it is needed.